# Gendered Citizenship, Inequality, and Well-Being: The Experience of Cross-National Families in Qatar during the Gulf Cooperation Council Crisis (2017–2021)

**DOI:** 10.3390/ijerph19116638

**Published:** 2022-05-29

**Authors:** Wahiba Abu-Ras, Khalid Elzamzamy, Maryam M. Burghul, Noora H. Al-Merri, Moumena Alajrad, Vardha A. Kharbanda

**Affiliations:** 1School of Social Work, Adelphi University, Garden City, NY 11530, USA; 2Institute of Living/Hartford HealthCare, Hartford, CT 06102, USA; kh.zamzamy@gmail.com; 3Doha Institute for Graduate Studies, Doha P.O. Box 200592, Qatar; maryam.mwafak.burghul@gmail.com; 4Supreme Judiciary Council, Doha P.O. Box 9673, Qatar; noura.almarri@hotmail.com; 5Center for Conflict and Humanitarian Studies (CHS)—Arab Center for Research and Policy Studies, Doha P.O. Box 200592, Qatar; moumena.alajrad@gmail.com; 6Department of Psychology, Adelphi University, Garden City, NY 11530, USA; vardhakharbanda@mail.adelphi.edu

**Keywords:** blockade, gendered citizenship, family cohesion, social inequality, cross-national families, family well-being, vulnerable children

## Abstract

This study explores the impact of gendered citizenship on the well-being of cross-national families following the political blockade imposed on Qatar in 2017. More specifically, it examines how these families, women, and children face challenges related to their lives, well-being, and rights. Twenty-three face-to-face interviews were conducted with Qatari and non-Qatari women and men married to non-Qatari spouses residing in Qatar. The study’s findings revealed that Qatari women with non-Qatari husbands did not enjoy the benefits of full citizenship, further undermining their psychological well-being and their socioeconomic and legal rights. Additionally, children born before or during the blockade have become stateless and undocumented, which jeopardizes their mental and physical well-being and the prospects of their parents’ economic advancement. This study contributes to the conceptualization of and debate on gender citizenship rules and policies, which can exclude these women and children and deny them the recognition and rights they deserve. Since ensuring full citizenship rights is crucial for people’s well-being, increasing gender equality and reforming Qatar’s existing citizenship policies would benefit both groups and provide social justice for all.

## 1. Introduction

On 5 June 2017, Bahrain, Egypt, Saudi Arabia, and the United Arab Emirates (UAE) imposed a land, air, and sea blockade on Qatar, claiming that Qatar supports terrorism, and ordered its diplomats and citizens to return home., The blockade has affected the sociopolitical and economic status of Qatar, which has a population of 2.69 million, for more than three years [1]. As of 25 April 2018, Qatari citizens living in the blockading countries have experienced 643 human rights violations regarding family reunification [2].

A substantial body of research shows that political conflicts or instability can affect people’s well-being [3,4]. The tangible effect could be the degradation of the affected country’s economic infrastructure and service delivery. Intangible effects could be, among others, a lack of trust in the government, a lack of social cohesion, the destruction of norms and values, citizen insecurity, and a negative outlook on the future. Women and children are the most vulnerable and prone to harm [5,6,7]. Gulf women married to citizens of countries within the Gulf Cooperation Council (GCC) that are involved in the conflict are disproportionately affected by the blockade and its aftermath [8]. In addition, the blockade has caused significant concerns for women living in GCC countries, including over the well-being of their children. The blockade has exposed how cross-national families are disproportionately impacted, may struggle to exercise citizenship rights and experience constraints on decisions regarding their country of residence. Thus, there is strong evidence that the lack of citizenship contributes to social, health, and economic disparity and discrimination [9]. Furthermore, research shows that exclusionary policies, such as limiting social rights and entitlements, lead to a growing rift between citizens and marginalized populations [10,11]. Therefore, some individuals experience difficulties in accessing social services due to citizenship exclusion policies and criteria [12].

Several studies have examined cross-national families and cross-national marriage in the Gulf region [13,14,15,16,17]. However, no systematic research has explored the effects of the blockade on the well-being of these families. This study aims to fill this knowledge gap, especially for women and children. It also provides a unique insight into gender-based citizenship and its impact on this specific group, especially during the current GCC crisis.

### 1.1. Cross-National Marriage and Citizenship Law in the State of Qatar

This paper defines cross-national marriage as “the union of two nationals from different countries and typically involves the migration of one member of the couple” [18] (p. 187). Such marriages may result in gender inequality and challenges in accessing full citizenship and residency rights, and they undermine national identity [19]. They may also influence cultural norms, masculine authority, and traditional citizenship laws and policies.

Cross-national marriages are common in the Middle East and the Gulf region. In Qatar, for instance, their rate increased from 16.5% to 21% between 1985 and 2015 [20]. The highest rate of Qatari women marrying men from GCC countries was in 1985 (9.9%) [13]. This kind of marriage resulted in one of the Qatari state’s first attempts to directly regulate family life. Additionally, the DIFI study showed that the greater instability of cross-national marriages resulted in a disproportionate number of divorces, affected Qatari women’s ability to transmit their nationality to their children, and made them less likely to marry foreigners. Qatari women believe that such marriages are cost-effective because their similar backgrounds are similar to those of people from other GCC countries and because of their exposure to different cultures [21]. However, the blockade has caused the number of cross-national marriages among Qatari women to decline from 124 to 19 [1,22,23]. Simultaneously, the divorce rate among these couples has increased from 25 cases in 2018 to 32 cases in 2019 [24]. Given the decline in the overall number of cross-national marriages, the blockade have may significantly affected these rates. Alternative explanations may be related to the Qatari government’s policy of providing incentives to newlywed Qataris, such as subsidizing wedding venues, or to the government’s direct intervention in the regulation of family life [25].

The dearth of relevant research makes it difficult to determine the exact number of such marriages in each GCC country and the real reason(s) for their increase or decrease rate. In addition, these marriages pose a threat to Qatar’s culture and heritage, since non-citizens outnumber citizens [26,27]. Nonetheless, Qatari women are beginning to realize how difficult it is to pass their citizenship on to their children.

### 1.2. State Law and Gender-Based Citizenship in Qatar: Heritage and Progress

Citizenship in most Arab countries is passed on blood and/or land. This law is controversial because it is based on a patriarchal system that prioritizes men over women [28]. Therefore, most Arab women married to foreigners cannot confer their nationality upon their children or husbands and often lose their original citizenship after marriage. The Gulf region has a complex citizenship hierarchy based on tribal, familial, economic, sectarian, and gender factors [29]. It gives the state exclusive jurisdiction to define citizenship and standardize it according to criteria and rights, responsibilities, and obligations. Beyond citizenship, rights are legal, social, political, economic, cultural, and intellectual practices. Consequently, these factors affect people’s access to rights and membership and shape their sense of belonging to communities, as well as their sense of self-worth and national identity [30]. The concept of citizenship also refers to people’s rights and freedom, their duties, and their expectation that the state will protect them from sudden changes by ensuring their health, thereby leading to higher civic participation. The state regulates citizenship by establishing eligibility policies to determine who may pass their citizenship on to their children or spouse and who may lose it [31]. 

Gendered citizenship refers to a differential relationship between the state and citizens based on gender, which leads to disparities in citizenship rights [32]. According to Chari [33], it has three aspects. The first aspect is the binary of the private and the public, with the public referring to material issues and the private to cultural matters. Second, the framing of rights within the social structures of caste, class, and ethnicity causes women to experience rights differently. Third, societal differences are conceptualized in light of multiple patriarchies. For example, according to Qatar’s Nationality law, Article 5, Act No. 38, of 2005, Qatari women “cannot confer nationality on their children under any circumstance and cannot confer nationality to their foreign spouses, if they are married to a non-citizen”. However, in Article 1.4 of Act No. 38 of 2005, Qatari men automatically confer nationality on their children and spouses, regardless of whether they are born abroad or in the country. The discourses on these two bodies of law intersect because marriage is one route to citizenship.

Qatar is also one of the eight most difficult countries in which to obtain citizenship. Law 38, Article 2, 2005 states that foreigners must be legal residents for 25 years before applying. Legally, divorced women can apply for a land loan only five years after their divorce. This reality has more than legal repercussions; it carries a social implication that negatively affects self-esteem and relationships [34]. In other words, the state’s law is the main force behind gendered citizenship. Consequently, denying women full citizenship status not only hinders their access to essential resources, but further impairs their well-being and that of their families [35].

Qatar is moving from a traditional to a modern society, forming new identities and redesigning conventional social roles. The High Commissioner for Human Rights report (2019) shows that the Qatari government actively empowers women, increases their leadership roles, and ensures their inclusion in the Shura Council. Women’s social roles in society are also changing, as more of them are attending colleges and working outside the home. Furthermore, they are more present in education, business, political leadership roles, and many other fields. “Nationalization”, also known as “Qatarization”, advocates the hiring of more local workers in the public and private sectors, thereby increasing opportunities for women to work in administrative roles and develop entrepreneurial skills [36]. The Qatari government also offers citizens many benefits: subsidized utilities, free healthcare, education, land grants, affordable housing, and an array of others [37]. Most recently, the children of Qatari women have begun to gain permanent residency rights due to a new law introduced during the blockade and implemented on 22 December 2020 [38]. This development represents an encouraging step toward ensuring the full citizenship rights of all women and children. However, Qatari women who marry foreigners still cannot pass on their citizenship to their children, a reality that further compromises gender equality and denies women full rights. 

### 1.3. Citizenship and Well-Being

One definition of citizenship holds that it is “a relationship between citizens and the government that is built on rights and obligations and the principle of inclusion and exclusion” [33] (p. 47). A newer definition stresses everyday functioning and emphasizes that political regulation alone cannot solve social issues. However, this discussion would benefit from academic input from the fields of, for example, sociology, public health, and family relations to explain and increase citizens’ well-being [39]. Citizenship is perceived as a privilege and a source of personal, national, and political identity [11]. Moreover, it provides rights and freedoms that promote a sense of self-worth and guarantees membership in a social group while holding people accountable for social, economic, and political decisions.

The World Health Organization defines well-being as referring to a state of “complete physical, mental, and social well-being, instead of just being free of disease or infirmity” [40]. Subjective well-being is conceptualized through three interrelated components: life satisfaction, pleasant affect, and unpleasant affect [41]. The term “affect” describes mood and emotions, whereas life satisfaction describes cognitive satisfaction with life [42]. One must possess positive emotion, engagement, positive relationships, meaning, and accomplishment to achieve positive well-being [43]. However, this concept has been widely debated due to its complexity. 

The objective approach covers “quality of life indicators such as material resources (e.g., income, food, housing) and social attributes (education, health, political voice, social networks and connections” [44] (p. 2). However, the definition and measurement of objective well-being have always posed a challenge. Therefore, it has been suggested that its dimensions should be explored rather than its definition [45]. 

As modern society has developed, a broader understanding of well-being has emerged around public health issues. State policies and practices are believed to impact people’s physical and mental health, particularly within families. In this vein, well-being is concerned with the objective and subjective aspects of individual well-being to inform policymakers and update international statistical indicators, such as the United Nations Development Programme’s Human Development Index [46,47,48]. Furthermore, some developed countries use well-being indicators to assess how state policies impact citizens’ well-being [49].

Demographic factors can also have a direct influence in this regard. In some countries, policymakers evaluate objective well-being based on such economic and wealth measures as, among others, household income, per capita production, and gross domestic product [50]. For instance, Qataris perceive socioeconomic status, religious affiliation, employment status, and trust in the government as its most essential determinants [51]. 

This paper argues that the concepts of citizenship and well-being have dimensions that overlap, intersect, or interconnect. The various definitions of citizenship and well-being illustrate that both concepts are complex and different in each society. To address this complexity and guide this study, we created a new conceptual multidimensional paradigm (see Figure 1) to illustrate this overlap using Leydet and White’s definitions of citizenship and well-being, respectively [52,53]. 

### 1.4. Perceived Interrelationship between Citizenship and Well-Being: A New Multidimensional Paradigm

This section illustrates the dimensions of citizenship and well-being and their interrelationship by using the arguments and definitions presented by Leydet and White. According to Leydet, citizenship consists of three dimensions: the law, politics, and identity. The legal dimension includes the civil, political, and social rights under which citizens have the freedom to enact laws and receive protection from them. The political dimension refers to citizens’ active role in a society’s political institutions. The identity dimension refers to citizenship as belonging (relational) to a specific group of people who share a common identity (e.g., national, political, social, and cultural). This dimension, also known as the psychological dimension, fosters belonging to one’s family, community, and society [52].

Similarly, White suggested three dimensions of well-being: subjective, relational, and material. The subjective dimension focuses on what people value and believe is right, along with their feelings and desires about themselves. It includes hope, fear, anxiety, satisfaction, trust, and confidence. The social or relational dimension includes relationships of love and care, social networks, and interactions with the state regarding law enforcement, local or national politics, social welfare, and security. The material dimension includes items commonly referred to as “human capital” or “abilities”, including income, wealth, employment, education, access to services, and the quality of the environment, as well as social capabilities, such as belonging to and inclusion in social groups and networks. In addition, it stresses the notions of trust, inclusion, membership, cohesion, and quality of life. It is determined by how people participate in their current and past social, cultural, and political lives [53].

## 2. Materials and Methods

This study was approved by Adelphi University’s Institutional Review Board. A semi-structured interview protocol was used to collect data from 23 participants via a set of predetermined questions asked in the same way to each participant. Questions included: What were the reactions to the blockade, opinions, and experiences? How did the event affect married life? How did it impact the children? We also asked several probing questions to explore gendered citizenship in depth, including its effect on women, legal rights, family cohesion, marriage, and access to services.

### 2.1. Data Analysis

Given this study’s exploratory nature and the general lack of relevant data and scientific research, we applied a grounded thematic approach to guide our analysis. Pilot coding (using Level 1 coding) to initiate codebook development began with two coders using 4 randomly selected interviews out of the total number of 23 interviews. To increase interpretative reliability, three research team members used three coding iterations: Level 1 coding was descriptive, identifying the prevalent general categories in the participants’ responses; Level 2 coding condensed the common groupings and used the first level of interpretive coding; and Level 3 coding identified significant themes and important implications. All coding levels were linked to Level 3 codes to help define themes and sub-themes within and across the data set. The research team considered different variables (e.g., gender, marital status, nationality, citizenship, and having school children) among participants to assess the blockade’s influence on their well-being and that of their families. All codes were classified and written separately to articulate contrasting points across the emerged categories. After the three researchers coded the themes individually, they met three times to review and identify new or additional themes and/or sub-themes, if any. Special codes and numbers were assigned to all participants. Qatari male and female participants were referred to as “QMP-#” and “QFP-#”, respectively. Non-Qatari male and female participants were referred to as “N-QMP-#” and “N-QFP-#”, respectively.

### 2.2. Ethics and Quality Assurance

To ensure quality, the research team randomly selected eight interviews from the transcribed interviews and compared the transcription with the recorded interviews before translation. The three team members then reexamined all translated interviews to ensure that the translation was accurate and contained the actual meaning of the content.

### 2.3. Participants

The study involved 23 participants (10 men and 13 women) married to spouses from Qatar and the blockading countries. Two-thirds of them (*n* = 15) identified as Qataris, four as Bahrainis, two as Emiratis, and two as Saudis. Their ages ranged from 30 to 54 years (M = 37.7), and they had between 1 and 6 children (M = 3). Nineteen participants reported earning bachelor’s or graduate degrees and holding full-time jobs. 

### 2.4. Procedure

Ten participants were recruited from a list of contacts collected from this project’s quantitative survey. Due to the topic’s political sensitivity, we used an additional snowball sampling method to recruit 13 more participants, all referred by colleagues.

Participants were included if they were (a) a Qatari citizen married or separated from a spouse from one of the blockading countries; (b) a Qatari citizen who was studying or had a business in one of the blockading countries; (c) currently residents of Qatar; and (d) 18 years or older and agreeing to participate. Upon meeting the study criteria, all participants were contacted again by phone or email and briefed twice about the study: before scheduling and when starting the interview. They read and signed a consent form attached to a letter describing the study’s purpose, potential benefits, risks, confidentiality issues, and voluntary participation. They were also asked verbally and in writing to consent to recording their interviews.

All but three participants were interviewed at their workplace. Each interview lasted 60 to 120 min and was conducted in Arabic by the principal investigator and one research team member. All interviews but one were audiotaped with a digital recorder, transcribed, and translated into English.

## 3. Results

### 3.1. Subjective Dimension

Psychological effects

The first sub-theme arising from the subjective dimension was the blockade’s psychological effects on the entire region and, more specifically, on the participants’ children and spouses. They all agreed that the blockade had negatively affected their well-being and left them with sad and painful experiences. One Qatari male described its effects on the entire region, stating:

The blockade is a black mark in the entire history of the Gulf region, not only for Qatar; the effects will stay for years to come. It is difficult to recover from what happened, mainly when everyone uses the media to express hatred and hostility (QMP-14).

A Qatari female participant expressed how it affected her psychological well-being:

I am anxious because I cannot reconcile with all those sitting around me. I have psychological stresses, and I am not my usual self. Even though I try not to be tragic, I can see my life is becoming darker and more tragic (QFP-2).

A Qatari male participant described the psychological impact on him and his relationship with others after the blockade, noting:

As a result of the psychological and financial pressure, I could not get out or socialize with others and my friends. The way I looked before was different than how I look now. My beard looked awful (QMP-10).

Family Separation, Fear, and Anxiety

Several participants expressed a wide range of negative emotions, including fear and anxiety of the unknown future and fear of family separation. A Qatari married female and two Qatari divorced males expressed concerns about family separation and their children’s future safety, as stated by one of the participants: 

Psychologically, I feel very fearful and anxious about the future of my children and husband. There is a chance they will be deported; even the idea of speaking about it frightens me. It makes me feel fearful, the same way when I am thinking about the future (QFP-3).

Similar concerns were expressed by a Qatari female participant about the future of her marriage and keeping her children in case her husband returned to his country:

I am always concerned and anxious about being separated from my children. If any problem occurs between my husband and me, it may end our marriage, and my husband will take the children and go back to Saudi Arabia. My children will be the victims (QFP-8).

A male participant expressed concern about being separated from his children, who lived with his wife:

I worry about my children growing up in their mother’s environment and changing their ideas and beliefs. They are still at an early stage where they are easily influenced. I am afraid of not being able to see them; I am worried they will completely change the way they think of me and Qatar (QMP-14).

Effects on Marital Relationships

Several Qatari men and women expressed concerns about their marital relationships. One of the extreme cases was described by a businessman who lost his business in Saudi Arabia. The situation had a severe impact on his relationship with his wife and children:

Due to the blockade, I lost all my investments in Saudi Arabia. My depression impacted my relationship with my wife and children; it got worse and deteriorated day after day. There were too many fights and tensions between us. In the end, she left the house and went back to her parent’s house, where she stayed for over two months. The marriage almost ended in divorce. 

Some identified their spouses as “other”, arguing they should not have full citizenship status. A Qatari divorced male, who was married to a non-Qatari woman, said: 

I am a father, born as a Qatari citizen and held the nationality of this state, but this is not the case with “the other person” [his former non-Qatari wife]. I do not mean to belittle the “other person”. I am saying this out of patriotism and loyalty; it is the sense of loyalty and belonging (QMP-12).

Another participant added:

I know people who are now married or engaged to spouses from Saudi Arabia. After the blockade, everything was ruined. People who marry within their country [nationality] are more likely to feel psychologically stable (QMP-14).

Regret and Opposition to Cross-National Marriages

The participants also associated sentimental feelings of pity and regret with citizenship. QFP-5 expressed regret about marrying a non-Qatari citizen, stating: “What I regret is that I got married to a man who is not of my nationality. But this is my fate”. Another participant, comparing her situation with that of her sister, shared: 

Wishing our situations are like my [other] sister, who is married to a Qatari man (QFP-5). My sister and I have the same problem [married to Saudis]. When we discuss our similar situation, we agree that “it was a wrong decision that we married to Saudi men”.

Even those who did not express such a regret wished for a better situation, as one of the Qatari female participants stated: 

About a week ago, my husband asked me: “Do you regret that you are married to me”? I told him: “No” … inside me, I feel I do not regret it, but I hoped for a better situation (QFP-6).

A Qatari male participant married to a non-Qatari female expressed his feeling that cross-national marriages are unstable:

I am married to a non-Qatari. Although I do not underestimate anyone or nationality, do not marry anyone outside Qatar because we do not know what the future holds (QMP-14).

Some of the participants who were married to Saudi women were entirely opposed to cross-national marriages:

Considering the time before the blockade, when I was still married to someone from another country, I had to travel two to three times a month between Qatar and Saudi Arabia. Speaking of the blockade now, I do not know what may happen in the future; if I marry a woman from a country other than Qatar, especially during a political problem. This is why I do not encourage such [cross-national] marriages (QMP-14).

Effects on Children

Children were negatively affected by the blockade’s aftermath, feeling terrified of the unknown and possible military attacks on Qatar, as stated by a Qatari male participant:

My daughter was afraid that blockading countries would start a war, invade and kill us. Her words reminded me of when I was young [during a similar situation]. I used to hear the teachers in school saying: “They will attack us, and they will destroy us” (QMP-15).

The children’s relationships with their relatives in GCC countries were negatively affected as well, as stated by a non-Qatari participant:

In the past, the children used to visit their aunts. I am afraid to say they hate them. My little child says, “We do not want to go to Bahrain anymore because they are blocking us” (N-QFP-20).

Effects on Extended Family Relations

A few participants also spoke of their extended family members’ ingroup conflicts and separation within cross-national households. 

A Qatari male spoke about his feeling of isolation since he has separated from his other family members and close relatives, who lived in all four blockading countries:

My mother is an Emirati. My aunts and uncles live in Bahrain and Saudi Arabia. Now, we live in isolation [even though] we used to have a great relationship. My brothers are married to my cousins [from Saudi Arabia], and my sisters are married to my Aunt’s sons [from Bahrain]. Because of the new blockading state rules, especially the “UAE Criminalization of Sympathy Law”, they are afraid to speak with us (QMP-15).

Another non-Qatari participant expressed sadness and anger because she could not see her father before he died: 

I have the same feeling of sadness and anger. I feel sadder over time when I remember that my father passed away, and I was not able to see him (QFP-21).

Several Qatari and non-Qatari participants expressed the fear that they would be stopped and separated from their extended families across the border. One participant described the challenges his Emirati wife faced:

My wife’s only concern is that she wants to see her brothers and sisters and find a way to do it, since she is an Emirati by origin. The problem is that she is Qatari now, so it is hard for her to see them! She must obtain a permit from the UAE to enter the country (QMP-15).

He also spoke of his relations with family members across the borders before and after the blockade:

It has become difficult for them [my extended family members] to call us because they are afraid; we are also scared to call them because they may suffer. Some people were hurt when they called their families; we heard that they were arrested and deported (QMP-15).

Another non-Qatari participant also expressed her grief over not being able to visit her family: 

We [non-Qatari women] are the most vulnerable because we got separated from our extended families, and we cannot travel to see them (N-QFP-18).

Effects on Social Relations: Isolation

Some of the participants verbalized their feelings of exclusion and isolation from their friends and society, both personally and virtually. One of them expressed her experience with her WhatsApp group:

My dealing with people has become more difficult since I began to avoid places where I would talk about the [blockade] crisis. There were WhatsApp groups I was part of, but I left every single one. I mean it. We do not talk about it. We do not have any discussion about the blockade. Even if someone mentions it, we act as if we did not hear it. Leaving the group was the best solution (N-QFP-18).

A Qatari female participant also described how the post-blockade environment affected her social relationships with others:

I have lost trust in people; it has faded away. I found it hard to maintain social contacts. Being with people was becoming less important. I began focusing more on my family, and I want to devote more of my life to them (QFP-3).

A third Qatari male expressed his anguish over not seeing his children after the blockade, which further isolated him from his surroundings:

This [blockade] has affected me and my work. My job is very demanding, and I must be awake and focused 200%. I tried not to engage with anyone and did not want to travel anywhere. Whenever I travel to a country, I stay in my hotel room. I felt my mind was preoccupied thinking about them, especially since there was no contact between us (QMP-14).

A non-Qatari male described his complete isolation, having once led a very active social life:

I never missed any activities in Qatar. I used to write, attend seminars, and post on my webpage. For example, I do not miss the Arab and International Relations forum. But now I have my laptop with me all the time. I stopped all activities (N-QMP-23).

### 3.2. Legal Dimension: Family Laws and Policies

Women’s Views about Citizenship: Gender Inequality

All the Qatari female participants verbalized their anguish over their unequal treatment. They expressed concerns at different levels, blaming their situation on Qatar’s citizenship and family laws and policies and, at the same time, demanding the same rights as their male counterparts. All the Qatari females and two neutralized male participants demanded equality by law, asserting that having the same legal citizenship rights is a right, not a privilege. One participant specified:

As a Qatari woman married to a non-Qatari man, I could not get housing. They [the officials] say a woman in Qatar should follow her husband. They tell me to take my kids and go back to my husband’s country. They cannot decide for me or deprive me of my right to choose. I am a human being. At last, it is up to me to decide, not them (QFP-3). 

Another female participant expressed skepticism about the equality concept and attributed it to the legal system:

The law talks about equality between men and women, but this equality means nothing; there is no real equality. 

Nearly all the female participants perceived Qatari laws as gendered and favoring males over females, whereas most of the male participants perceived Qatari laws as favoring women. A Qatari female participant stated:

The law allows the naturalization of the wife of a Qatari man; why don’t you do the same for the husband of a Qatari woman? The government keeps talking about families and celebrating the ‘Family Day’ and ‘Family Welfare’. So why does the law differentiate between a Qatari man married to a Saudi woman and a Qatari woman married to a Saudi man? So we are not equal (QFP-8).

Another woman questioned the double standards of the state’s family laws: 

We tried to solve Qatari women’s concerns about citizenship. However, officials told us that this [granting citizenship to their children and spouses] can cause “genealogical mixing”. So Qatari men can marry Egyptian, Indian, Bengali, Moroccan, or Syrian women. Also, these women can bring their parents here and decide who they [their children] should marry. “Why is it only considered a genealogical mix if the woman is Qatari”? (QFP-4).

A third participant criticized the new government’s initiative to grant permanent residency to the children of Qatari mothers by saying: 

When the issue of permanent residency came up, I felt as if the Council was telling us: “You either get it or forget [the passport]”. Permanent residence law is similar to a valid visa. Granting children permanent residency would mean muting us. My children are still young; they do not work or attend university. We long for a decision to naturalize Qatari women’s children (QFP-5).

Men and Gendered Citizenship: Women are More Privileged than Men

Some of the male participants felt envious of women for being advantaged over men, as described by a male participant:

I swear I would not stay under my husband’s authority if I were a woman. I would rather be divorced because he [my husband] will pay me monthly. I would be an idiot if I agreed to remain as his wife. Women enjoy maternity hours. They have off days more than men and have more rights than the law grants men (QMP-17).

Another participant supported the same belief:

Society is striving to give women their rights. When a woman leaves her house, the judge makes her husband pay her dowry, including the postponed dowry and alimony. She [a Qatari woman] can force her husband to leave the house, but he cannot evict her. So why should she be equal to me? (QMP-12).

Another Qatari male participant viewed Qatar’s family laws as destructive to family unity and cohesion and believed that Qatari laws favored women over men:

These laws promote differences in perceptions of equality and create conflicts within families while giving women more power over men. Even if such a [non-Qatari] person has spent all their lives in Qatar, they still have no relatives. I may offer this person 40% of citizenship privileges, but not all citizenship rights. Therefore, I can never deal with this person as an equal citizen (QMP-17).

Some of the participants believed that gender inequality is derived from tradition and social norms that obligated men to provide their wives or ex-wives with necessities, as explained by a non-Qatari male participant married to a Qatari woman:

Even after divorce, she [his Qatari wife] receives her basic needs such as food. “When fathers tell their daughters to get divorced and return to their country, you will probably understand why divorces are so common today” (N-QMP-23).

A few male participants believed that the country’s social and traditional values are the reason for the disparity between men and women, as stated by one Qatari male participant: 

I believe the Qatari woman has her rights in society, but the society oppresses her with traditional and family values (QMP-12). 

Another participant thought that Qatari society continues to treat men and women unequally, even after separation: 

Although I divorced my wife, the law required me to provide travel permission to my ex-wife and daughter when they travel to Egypt (QMP-13).

Effects on Children’s Well-being: The Invisible Citizens

Citizenship issues also created anxiety among the participants, mostly about accessing the country’s resources and services. All but one of the women married to non-Qataris expressed concern over their children’s future in this regard. Their inability to pass on their citizenship increased their sense of insecurity and prevented them from accessing the country’s resources. However, the anxiety levels and their persistence varied according to gender, citizenship status, and the source of concern. For example, the Qatari women married to foreigners were more affected than the non-Qatari female and male participants, as stated by a Qatari female participant: “No one was harmed by the blockade as badly as I was and the other Qatari women like me” (QFP-8). 

According to a Qatari female and a non-Qatari male participant, children struggle to preserve general well-being and prosperity if their basic needs are not met. The participants also referred to their children as “invisible citizens” and discussed their concerns about their futures and their ability to obtain essential services. They elaborated on Qatar’s family laws, claiming that administrative difficulties degraded their children in exactly this way.

What should I do? My husband is out of work; my son is living in the State of Qatar with no I.D. proof. My son is not in the system, he does not virtually exist in terms of his vaccinations and sickness, and he is about to be two years old. My daughter’s passport has expired, and my other daughter’s passport expires next year in September (2020). What shall I do? (QFP-1). 

Another non-Qatari participant described how his children were victimized by the political tensions between Qatar and its neighbors. 

I am a resident of Doha. My wife is Qatari. The government issued a ruling about not giving my girl a passport due to the political crisis between Qatar and the UAE. It was a clear legal text. The [GCC] crisis is not my fault; when there is a political problem between the two countries, what do the cross-national and Emirati families do with that? 

The same participant further explained that these rules apply in the whole Gulf region, including his country, elaborating:

According to Emirati Law, the parents can demand citizenship for their child [if born outside the country] if they are under three years old; otherwise, they will be denied citizenship. Thus, I cannot demand an I.D. for my daughter (N-QMP-23).

During the blockade, health services and resources for children without legal documentation were severely limited. Two Qatari female participants expressed concern about their children’s right to health care and education:

When I took [my daughter] to the hospital, she was considered a foreigner, and I should pay (her medical costs). She has no health insurance I.D. Thus, I had to take her to a private hospital. As a baby, she received a vaccination book. She does not have any other proof (QFP-2).

The same participant expressed concern about her son not receiving an education:

My son cannot receive education; I went to private kindergartens, but they sought his birth certificate. The one I had is unofficial. They demanded a passport or I.D. card to prove [that] the child does exist; what can I do? (QFP-2).

Another participant commented on the significance of having citizenship before and after the blockade:

Before the blockade, I did not ask for citizenship for the children. My only concern now is passing citizenship on to my children so that they will feel settled and comfortable (QFP-4).

The participants expressed concerns about their children’s future, whether they could find a job and go to college, and whether they would be treated exactly like other Qatari children. One participant shared her concerns over her daughter’s future: 

My daughter has the Qatari document [residency]. However, she will have second priority when applying for a job because her mother is a Qatari married to a non-Qatari (QFP-4). 

### 3.3. Resources and Material Dimension

Limited access to land and loans

Two of the naturalized Qatari male participants criticized Qatar’s policies regarding land ownership. One stated: 

I believe that this law is biased. As far as land goes, I do not see equality. Sometimes, when I seek my rights, they tell me that I am naturalized. Due to this law, my father and I could not get citizenship (QMP-13).

A second participant added: “Real estate is something I can invest in with limits; the priority is for the citizens of Qatar” (QMP-17).

Another female reflected on her limited options regarding her ownership of land: “They [the government] never give us a piece of land or a loan; they hardly care for us” (QFP-5).

The Experience of Non-Qatari women married to Qatari men

Despite being forced to give up their nationality or citizenship to marry Qataris, four non-Qatari females showed their appreciation of, loyalty and gratitude to, and pride in the Qatari people and government for granting them citizenship and treating them as Qataris. One participant syayed:

I have the right to access health care at hospitals. I have been treated the same as a Qatari. I joined Qatar University without fees, the same as a Qatari citizen. After graduation, I had a work interview with a Qatari woman. I was employed. Frankly, there was no discrimination (N-QFP-18).

Non-Qatari women married to Qatari men had a choice: either keep their citizenship or claim their husbands. In either way, their children were entitled to their father’s citizenship. However, all the non-Qatari married participants in this study chose Qatari citizenship for the sake of their children’s future, as expressed by a non-Qatari female participant:

When he brought me the [citizenship] papers to sign, I cried that day as if I [had] lost both of my parents. It is tough to abandon your citizenship. When I went to Bahrain with the Qatari passport, my uncle asked me: “Did you dump your country”? I won’t forget his words! I replied, “No, I did not, but it was for the sake of my children; I do not want them to be unemployed. If I had kept my Bahraini nationality, I would not have enrolled in the university and got a Qatari citizen’s privileges. I would not have been employed because priority is always given to Qataris. I have a son in the military corps. If his mother were not a Qatari citizen, he would not have joined this corps. It has been a blessing. I am now reaping the fruits of the decision I took 26 years ago” (N-QFP-21).

### 3.4. Identity Dimension 

Identity Crisis

All of the Qatari women and some of the divorced Qatari men reported that their children questioned their identities, which either strengthened their national identities or caused identity crises, confusion, and unrest within themselves and their relationships. Some children born to non- Qatari fathers felt inferior, which engendered confusion and anxiety. A Qatari female participant described her three-year-old son’s confusion in this regard:

Why don’t I become a Qatari? When I heard my son asking this question, I was shocked and asked him: “Why do you want to become a Qatari? What is wrong with being a Bahraini”? He replied: “Because they [the Bahrainis] are among the blockading countries, and I do not want to belong to one of these countries because, at school, kids say that we belong to the blockading countries. When will this situation end”? He still does not know how to pronounce BLOCKADE (حصار) correctly or differentiate between it and other similar words in Arabic, i.e., STORM إعصار))? (QFP-4).

Another Qatari female married to an Egyptian man described her son’s identity crisis while trying to fit into the Qatari identity:

Lately, I felt he [my son] had no pride in his Egyptian identity. He used to like that before the blockade, but he did not like it anymore. I think he loves to speak more Qatari dialect, even how he says. Now, he wants to mingle with the Qatari people. I think he was trying to forget about being Egyptian and avoiding talking to someone with the Egyptian dialect (QFP-3).

Another Qatari female participant described her nine-year-old son’s fear about revealing his true national identity to his classmates and friends by claiming the Qatari identity instead:

My nine-year-old son told me: “In school, they say: ‘You are from the countries of the blockade”. Why do they tell us that we are from the countries of the blockade”? As a precaution, I told him to tell his classmates that he is from Qatar, lives in Qatar, and does not know anything about these countries. The next day, I asked him: “Did you tell them that you are a Qatari and live in Qatar”? He answered, “Yes. They even said, ‘Why didn’t you tell us before that you are a Qatari living in Qatar? We thought you were from the blockading countries”! He felt good that he could deceive them (QFP-4).

Similarly, a Qatari female participant reported that her daughter rejected her original Saudi nationality and told her: “I am a Qatari”. She responded: “Okay, you are a Qatari, but your father is Saudi, so you are of Saudi origin”. Her daughter replied: “No, I do not want my father to be Saudi”. The daughter wanted her father become Qatari (QFP-5).

The children of Qatari fathers and non-Qatari mothers developed a more robust sense of national identity. By contrast, the children of Qatari mothers and non-Qatari fathers were more likely to struggle with national identity confusion. As stated by a Qatari male participant:

I noticed that my daughter’s sense of belonging to Qatar strengthened after the blockade. Suddenly, she talked about politics and told us what we should do and what Shaikha Mouza (Her Highness) and Shaikh Tamim did or said. In a sense, it is like how we all talk about safeguarding Qatar (QMP-15). 

## 4. Discussion

This study contributes to the literature on traumatic events and helps people to understand how these events may affect lives, especially those of women and children in cross-national marriages. The suggested multidimensional conceptual model may also contribute to our understanding of the perceived link between well-being and gendered citizenship, as shown in Figure 1. The findings highlight how women’s citizenship is gendered by the state’s legal policies, which affect all aspects of their lives, including issues relating to subjectivity, relationships, the law, politics, material life, and identity. The findings also indicate how Qatar’s socio-cultural and patriarchal values influence women’s experience of gendered citizenship. The results show that the quest for citizenship among Qatari and non-Qatari spouses continues to involve legal and social inequality, as it does for other women in the Gulf and the larger Middle East.

The multidimensional model (Figure 1) illustrates the interconnectedness and interdependence of all six dimensions. Deficiencies in one of these dimensions automatically affect the others. To attain a good standard of well-being, citizens need to enjoy access basic needs, maintain social connections with others, integrate into their communities, have a social and national identity, and feel that they belong. Furthermore, they must have the right to access material resources, including employment, income, education, housing, and health care. They also need to feel protected by the state’s laws and policies and fully engage in the political process to enhance their families, communities, and sense of social integration. In sum, individuals who feel valued are more likely to be socially involved, foster a sense of belonging, trust their government, fulfill their obligations, develop a national identity, and lead more positive and healthier lives [54,55,56,57].

Because citizenship and well-being are strongly linked to social, economic, political, and cultural dimensions, Qatari policymakers, researchers, and practitioners must consider the cultural contexts through which citizens perceive their well-being. For example, in collectivist cultures such as Qatar, perceived happiness is linked to the well-being of family, friends, the community, and society rather than that of the individual [58]. Conversely, in Western societies, happiness is achieved through the social recognition of a person’s worth and the accomplishment of goals [59]. Therefore, gendered citizenship encompasses more than just the perceived relationship between citizen and state. It considers traditional institutions, such as cultural and religious values, the state’s family laws and policies, the household, civil society organizations, the economy, the political climate, and other elements of social life that affect both genders’ equality, equity, and rights.

This study’s results indicate the political dimension (blockade)’s negative impact on Qataris and non-Qataris, especially on women in cross-national families. During this political crisis, family and citizenship laws (the legal dimension) deepened the adverse effects on these families. These laws’ unique socio-cultural, religious, and traditional gender-based contexts (the social/relational dimension) further excluded them from the process, thereby undermining their well-being and marginalizing their social and national identities (the identity dimension). Even though gender-based citizenship issues are common in the Middle East, the treatment of women and children in cross-national marriages and during the blockade significantly affected their quality of life and living conditions (the subjective/psychological dimension). Another critical issue is the lack of family and community cohesion. These participants were deeply concerned about family separation and the possible deportation of their non-Qatari spouses and children, as well as separation from their extended families. 

The results of this study are similar to those of previous research, which showed that families experiencing similar circumstances face multiple psychosocial issues, including financial hardship, housing problems, and food insecurity [60,61,62,63]. Such political crises negatively affect children’s well-being due to family fragmentation, conflict, and separation [64,65]. According to scholars, the fear of separation influences family relationships and cohesion. For instance, Victor found a strong association between parental mental health and family cohesion [66]. Family cohesion, or its absence, is also associated with both positive and negative mental health outcomes [67]. Families that lack cohesion or are threatened with separation may experience feelings of abandonment [68]. Due to forced or sudden deportation, family separation can negatively affect children’s psychological and academic development [69,70]. Its effects persist even after families reunite [60,71]. Furthermore, the failure to honor and include one family member can negatively affect the entire family’s cohesion and worsen the psychological distress of its members [72].

Gendered citizenship appears to have exacerbated psychological distress in Qatar’s cross-national families. Furthermore, cross-national marriages tend to be less stable than those between spouses of the same nationality [20,73]. The lack of family stability contributed further to the participants’ social exclusion. Anguish and psychological distress are often attributed to women and children’s social and national identities, suggesting that women and men attach different sentiments to their citizenship. Their perceptions involve recognition, the acknowledgment of women as equal partners, equal rights, visibility, inclusion, and engagement, in addition to access to material benefits. The loss of citizenship caused the participants to feel rejected, alienated, excluded, and marginalized. In addition, these women often felt anxious and isolated, objectified, and unstable because they are excluded from their families, friends, local communities, and social groups [74,75]. Their children’s social invisibility only added to their alienation. This invisibility and “otherness” increased the women’s anxiety about whether their children would receive proper health care and education or find employment when they grew up.

Children born to cross-national families may experience long-term inequality and low societal mobility, low academic achievement, and unequal access to employment [76]. Families with undocumented members are also at high risk of poverty, which affects their trajectory [77]. This lack of equality may adversely affect their ability to construct social or national identities, further reducing their sense of belonging and ability to form relationships [78,79]. In support of our results, Browne et al. found that Arab immigrant children experienced the highest amount of emotional distress over time compared to other minority groups. This distress could be due to the high value that Arab culture places on family and friends.

The interaction between law, tradition, family life, and well-being is complex and multidimensional. Gendered citizenship is fundamental to this interplay. While citizenship contributes to positive psychological well-being, engagement, and interpersonal interactions, gender-based citizenship places women at a disadvantage and makes them more vulnerable to psychological distress [11,80,81].

According to the multidimensional model, examining well-being within legal contexts may allow us to understand the normative effect of political and governmental intervention on the lives of cross-national families. This model combines multiple dimensions with a reasonable consideration of cultural and national contexts, especially the Middle East’s collectivist cultures. Cultural values and socialization profoundly shape gender hierarchies and relationships. In the Middle East and North Africa (MENA), gender is framed around the male–female binary: the husband is perceived as the head of the household, the protector, and the breadwinner, whereas the wife is the mother and caregiver. Understanding well-being should therefore be based on individuals’ perceptions of their lives, such as their daily activities, family and social relationships, and life aspirations, as well as traditional and modern social norms. All these dimensions coincide and impact one another positively and negatively, depending on the political climate and degree of governmental interference. Furthermore, because citizens are not equally involved in civic, social, and political activities, their social situations may differ depending on their gender, socioeconomic status, and educational level [56]. Consequently, people’s positive experiences with each of the citizenship dimensions (legal, political, and identity) improve their overall welfare and satisfaction and, in turn, positively affect their subjective, relational, and material well-being.

### Limitations

Although this was the first study to address gendered citizenship and well-being during the blockade, it has a few limitations. First, the study’s use of a small sample, based on a qualitative method and focusing on cross-national families, mean that it cannot be generalized to the entire Qatari or Gulf population. Furthermore, the study focuses on the meaning the participants ascribed to their experiences, rather than verifying those experiences by using reliable and validated instruments and measures. As a result, our interpretation and discussion of our findings could be biased. Future research could address similar countries in the region, especially those involved in the political crisis, by using a mixed-method design and a larger sample size, including cross-national and national families.

## 5. Conclusions and Implications

This study contributes conceptually and in terms of knowledge to the notion that societal principles of gender equality profoundly influence citizenship rights and well-being outcomes. Guided by the multidimensional model, these results show how Qatari families perceived their well-being during the Gulf political crisis and how citizenship affected their well-being. As illustrated in Figure 1, the multidimensional model incorporates well-being’s subjective and objective elements, suggesting a connection between perceived citizenship and well-being and how each of these dimensions affects women in cross-national marriages. In addition, cultural and traditional norms and sociopolitical and structural factors shape gendered citizenship in the Gulf region. Naturalizing the children and spouses of Qatari citizens married to foreigners contributes to their sense of belonging, fosters family cohesion, increases their integration into society, and improves their quality of life and happiness. Social and family well-being occurs in tandem with positive individual well-being. Citizens’ subjective and objective well-being assumes that “life quality, comfort, happiness, security, safety always posit humans as both recipients and agents” [82].

The multidimensional model clearly explains this study’s results and explores how life events, such as the blockade, affect people in cross-national marriages and their well-being at the micro, mezzo, and macro levels. Based on this model, citizenship transcends rights and responsibilities. It is a lived experience whose meaning differs depending on a given society’s cultural and political contexts.

Additionally, citizenship is constructed differently depending on other sociodemographic factors, such as gender, religion, age, nationality, marital status, and class. Qatari residents of all ages suffered as a consequence of the blockade. However, the women and children in cross-national marriages suffered the most. Qatar and other Gulf states must reform their laws governing citizenship and family life. Policymakers need to identify the connection between gendered citizenship and the well-being of individuals, families, communities, and society. They can then use their findings to determine how to implement new policies that effectively limit the adverse impact of gendered citizenship and better understand its impact on the lives of women and their families. Qatar’s 2030 National Vision and National Development Strategy should reconsider its 2030 agenda and grant women married to non-citizens and their children full citizenship rights as a major priority to avoid the negative consequences described above. These women must be documented as citizens with full rights and as integral members of Qatari society, rather than as men’s property or as foreigners in an adopted country.

Education can provide the necessary changes. Education based on equal citizenship is one way to address the region’s political and social realities. Furthermore, the curriculum must be updated to include discussions of national identity, women’s participation in and contribution to government and politics, and the transmission of democratic and just values to future generations. This resocialization process would enable educators and parents to contribute to gender equality, social justice, and the advancement of human rights.

At the practice level, health care, mental health, and social and legal professional providers must understand the socio-cultural and political context of women, especially those in cross-national marriages, and the impact of gender citizenship on their family’s well-being. The mental health and social providers who serve these people must reassess and redesign their services accordingly. Particular attention must be paid to how family cohesion reduces the exposure of children and adolescents, especially those who are marginalized, to violence and mental illness because of their second-class citizenship. This study also emphasizes the interrelationship between subjective and objective well-being and women’s citizenship as a form of inequality. The government is responsible for ensuring that all Qatari citizens, including those married to non-Qataris, their children, and their spouses who reside in Qatar, have access to all public services and resources, including health care, education, housing, employment, and income.

Qatar is home to a wide range of civil-society organizations that support vulnerable people and cultural outlets for its many different nationalities. These organizations can also actively assess and address current human rights concerns and increase women’s civic involvement locally and nationally. In addition, informal, but influential, organizations can offer workshops and training sessions designed to strengthen women’s sense of citizenship. Workshops may explore economic and social access, gendered citizenship, cross-national marriages, and issues facing women in the workplace or political arena. The goal of this education should transcend educating women and marginalized groups to teach policymakers how to become inclusive and practice justice toward women, as equal citizens, by incorporating all six dimensions of citizenship and well-being.

To conclude, the multidimensional model suggests that gendered citizenship accentuates well-being and gender inequalities. It also views citizenship and well-being as major domains that hinder or enhance well-being. Therefore, special attention must be paid to the contexts of individuals, families, and groups with marginalized identities and statuses when developing relevant programs and services. Maintaining a healthy environment and society involves paying attention to family cohesion and structures, opportunities for growth, and the betterment of individual prospects.

## Figures and Tables

**Figure 1 ijerph-19-06638-f001:**
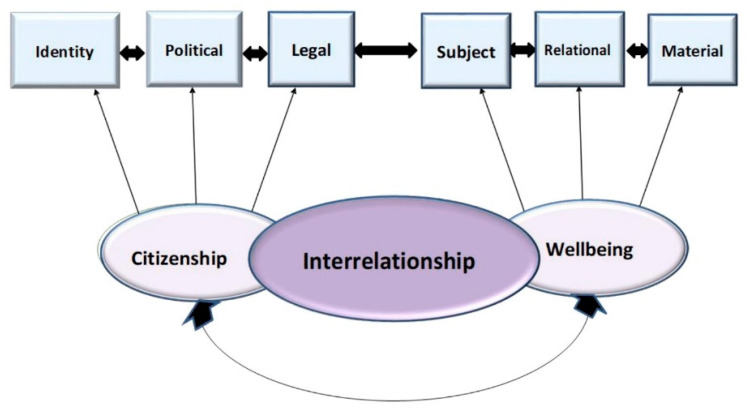
Multidimensional Model.

## Data Availability

The data are not publicly available due to the Ethical Committee Institution’s restrictions.

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
