# Peer review of "Gendered Citizenship, Inequality, and Well-Being: The Experience of Cross-National Families in Qatar during the Gulf Cooperation Council Crisis (2017–2021)"

_ijerph, 2022, doi:10.3390/ijerph19116638_

Round 1

Reviewer 1 Report

This research is a qualitative interview survey. The focus of the research is the controversial issue of women’s and children’s education. It pointed out that international marriage is of substantial help to Qatari women. The research examines how regions of the world with hostile political climates can make women happy. The results can give the world an international perspective on how to help Qatari women and children.

Its findings also show that Qatari society has valued promoting family cohesion. The 23 subjects interviewed, it has shown expectations.

Author Response

 Dear Reviewer,

Thank you so much for reviewing our manuscript. We truly appreciate your time, efforts, and support of this manuscript. To address your suggestion about improving the methodology, we made all efforts to describe exactly what we process step by step unless you have something more specific to address.

Thank you again 

Reviewer 2 Report

Please see word document attached

Author Response

Dear Reviewer,

My team and I wish to thank you for your time, efforts, and support in reviewing our manuscript.

Your thorough review and comments are very much appreciated. 

Attached is our reply to all your comments. We made all efforts to address them all as suggested.  All comments were accepted and corrections were made to all.

This manuscript is a resubmission of an earlier submission. The following is a list of the peer review reports and author responses from that submission.

Round 1

Reviewer 1 Report

The present study is evaluating the impact of gendered citizenship on health and well-being during the GCC crisis (2017-2021). In the qualitative study, the authors interviewed 23 males and females participants under three hierarchical contents with each level. The readers would understand the situation on those who live in Qatar. In addition, the gendered citizenship on Safety levels were also assessed. Although the importance of the present issue, there are several important point to correct.

Major Comment

1. The contents and structure of the present study may be not proper the present journal. This paper applied for article type of IJERPH. However, the contents of this paper seems not concise and descriptive due to its study methods. In addition, it seems vary lengthy such as official reports. This means that the authors should make the article into summarised to show the proper priori of the study hypothesis. 

2. The reference style in main draft and reference section was not following the style of the IJERPH. This is essential component of submission. The author should consider the author guideline.

3. The contents might be related with social structure and personal relationship. This issue might be proper for the Jouranls concerning on social issues or any other safety issue rather than health journal. The authors made the title as impact Gendered Citizenship on health and well-being however, those impacts were not reasoned by the interview. In the discussion, implement and conclusion section, the authors described the health effects by the current situation, however, in the results section, just the current situation was described without the current health problems by interview. Those makes this article feel vague. 

Reviewer 2 Report

The manuscript deals with transnational families in Qatar and the challenges they experience since the blockade. Specifically, the authors examine the impact of citizenship laws on Qatari women and children's well-being, living with their transnational family members, and the legal disparities within the citizenship laws for males and females, during the blockade.

Given the disparity between female and male citizenship rights in Qatar, Qatar is indeed a very relevant case study to untangle gendered citizenship within a crisis context. This is why the topic is relevant to scientific community as well as beyond academia and brings important insights. Since it will be published in open access, it has the potential to become reference for university students as well as international think tanks.

I do not have a psychology background and so the comments I make are coming from a member of scientific community but from a different discipline.

In its current version, the article’s main point that can be improved is the overall structure and the way data is presented. Some minor comments follow but this is a key element.

First, the overall structure shall be made clearer, in the current version, the hierarchy between headings is not clear which complicates the understanding by the reader. Second, a clear announcement of the structure of the paper in the introduction would help the reader to know where the article is going.

While the introduction, sections on methodology, discussion and concluding remarks are rather clear, the section “findings” is confusing and often times the data presented do not clearly relate to the overall argument of the article.

In my opinion, this is due to the lack of distinction within the sample: Qatari women, non-Qatari women, Qatari men and non-Qatari men are all put together in a melting pot section “Findings” with quotes from one category meeting the quotes of others with no clear link between them or no clear line of thought exemplified by the authors. The section “he said, she said” is ethically problematic with quotes by privileged Qatari men contrasted with experiences of discrimination by non-Qatari men and Qatari women married to non-Qatari men with non-Qatari women married to Qatari men. The section should analytically categorize the main findings within each category, then make them speak together (while still keeping the high ethical standard of not putting on the same footing someone speaking with the position of power with someone suffering from discrimination) and always relate the finding to the overall argument of the article – which is – what does it tell about the gendered citizenship in Qatar.

The focus on Qatari and non-Qatari men in justified in a sense because gender cannot be equated with women but the article explicitly focuses on “women and children” so the place accorded to men is less justified.

Finally, the manuscript doesn’t address temporality adequately well. It says to focus on the blockade (post 2017 period) but it is clear that the discrimination of Qatari women married to non-Qatari men is structural and of longer date. These two temporal layers shall be recognized and distinguished.    

Minor points:

Line 87: define wellbeing, it is a central concept to the article and thus shall be defined.

Line 105: “Since the blockade, the rate of Qatari women married to non-Qatari men from the GCC countries went down from 124 in 2017 to only 19 cases in 2020” – does this number relate to the number of marriages contracted that year or of transnational marriages existing overall?

Line 121: “…England left the European Union in 2016 to regain control over its immigration and borders. Similarly, in the USA, most Americans elected Trump in 2016 based on his anti-migration policies (Bryceson, 2019).“ both of these points are problematic and there is no agreement within the scholarly community, for instance, that UK left EU to regain control over its immigration.

Line 200: „Despite 200 all these efforts towards modernization, Qatar has not cut ties with its past; instead, it focuses on creating a society where tradition meets modernity (Golkowska, 2014).“ I am not familiar with the reference, but the sentence doesn’t seem to have academic rigor. It is also not clear what it would truly mean.

Line 215: What is “female participation in law”?

Line 222: “Since men outnumber Qatari women, nearly two to one, men hold the most powerful positions, leaving women underrepresented in most of the life spheres (Breslin & Jones, 2010).“ the issue of outnumbering is indeed very relevant here, but even if it wasn't the case women would still be largely underrepresented

Line 235: “Since Qatari citizens are a minority in their own country, they see transnational marriages as a threat to their cultural lineage by foreign influences (Babar, 2015; Davidson, 2008).” Either the reference provides clear opinion polls or it is not a scientific claim. Qatari women married to non-Qatari men probably do not see translational marriages as such a threat. It may be better to talk about what the governmental policies in this regard are and what it tends to justify them with.

Line 264: requirements met by study participants: meeting all of the requirements is probably not possible (1 and 4 are mutually exclusive), while all of the requirements should be coupled with requirement n°5. it should be made clearer.

Line 381: the participants should be either assigned codes (for instance, FP1 - for female participant 1) or given pseudonyms, otherwise, it is not possible for the reader to evaluate whether the same person has been quoted multiple times or whether the examples draw systematically on different people's responses.

Starting from line 403, that section is completely unrelated to the paragraph

Line 522: “Even with citizenship and civic rights like divorce, women still are held as inherently unequal to men, as stated by a Qatari naturalized male participant who was informed that his former wife and daughter were traveling without his permission“

How is this related to well-being? Authors do not link this man’s statement to the overall question. Does it mean that his testimony tells something about the lack of well-being of the women under his guardianship?

Line 604: This should be a starting point of a section which exemplifies what changed after the blockade: “Citizenship provides individuals with security to navigate the country's systems. In their views, without citizenship, the individual is left with barriers and obstacles rather than access and opportunities. After the blockade, the value of citizenship has further increased. A Qatari female participant emphasized the seriousness of citizenship before and after the embargo.”

Lines 671-677 section about QNV has no connection to the argument of the article what so ever.

Lines 878-881: as final lines of the manuscript, they are empty of sense with regards to the overall argument and seem cut out of context.

Finally, the overall relevance of the article should be exemplified: what does it show about gendered citizenship beyond the Qatar’s case?  

Reviewer 3 Report

Good work on an interesting topic. The entire manuscript needs more work to make it more cohesive and not just an extensive literature review. The results presented are easily lost as this manuscript looks more as a literature review. 

Reviewer 4 Report

1. To my mind, it is not enough 23 interviews. 

2. .....when England left the European Union in 2016 .... Is this correct? Please, check.

3. Figure 1. - there are 4 different pictures. It is rather confusing the reader. Try clear it. 

4. Implications and Conclusion 

As a rule, it is supposed to be mostly based on the results, findings of the research. 

5. Please, check references. For example, in session References : Bryceson, D. F. (2019).....  is written twice